# Preparation and Characterisation of Polyphenol-HP-β-Cyclodextrin Inclusion Complex that Protects Lamb Tripe Protein against Oxidation

**DOI:** 10.3390/molecules24244487

**Published:** 2019-12-07

**Authors:** Wenhui Li, Lidan Ran, Fei Liu, Ran Hou, Wei Zhao, Yingbiao Li, Chunyan Wang, Juan Dong

**Affiliations:** 1School of Food Science and Technology, Shihezi University, Shihezi 832000, China; liwenhui_mm@126.com (W.L.); m13150400832@163.com (L.R.); 18899592269@163.com (R.H.); lybfood@sohu.com (Y.L.); wcy1510016681@sohu.com (C.W.); 2College of Life and Geography science Kashgar University, Kashi 844006, Xinjiang, China; 18040836705@163.com; 3College of Food, Jiangnan University, Wuxi 214122, China; zhaow@jiangnan.edu.cn

**Keywords:** grape seed extract, inclusion complex, hydroxypropyl cyclodextrin, antioxidant, myofibrillar protein, oxidation

## Abstract

Grape seed extract (GSE) displays strong antioxidant activity, but its instability creates barriers to its applications. Herein, three HP-β-CD/GSE inclusion complexes with host–guest ratios of 1:0.5, 1:1, and 1:2 were successfully prepared by co-precipitation method to improve stability. Successful embedding of GSE in the HP-β-CD cavity was confirmed by fourier-transform infrared spectroscopy (FTIR), X-ray diffraction (XRD), differential scanning calorimetry (DSC), and scanning electron microscopy (SEM) analyses. The Autodock Tools 1.5.6 was used to simulate the three-dimensional supramolecular structure of the inclusion complex of 2-hydroxypropyl-β-cyclodextrin and grape seed extract (HP-β-CD/GSE) by molecular docking. The MALDI-TOF-MS technology and chemical database Pubchem, and structural database PDB were combined to reconstitute the three-dimensional structure of target protein. The binding mode of the HP-β-CD/GSE inclusion complex to target protein was studied at the molecular level, and the antioxidant ability of the resulting HP-β-CD/GSE inclusion complexes was investigated by measuring 2,2-diphenyl-1-picrylhydrazyl (DPPH) free radical scavenging. The effects of HP-β-CD/GSE on myofibrillar protein from lamb tripe were also investigated under oxidative conditions. The positions and interactions of the binding sites of HP-β-CD/GSE inclusion complexes and target protein receptors were simulated by molecular docking. The results showed that HP-β-CD/GSE inclusion complexes were successfully prepared, optimally at a molar ratio of 1:2. At low (5 μmol/g) to medium (105 μmol/g) concentrations, HP-β-CD/GSE inclusion complexes decreased the carbonyl content, hydrophobicity, and protein aggregation of myofibrillar protein from lamb tripe, and increased the sulphydryl content. Furthermore, high concentration (155 μmol/g) of HP-β-CD/GSE inclusion complexes promoted protein oxidation.

## 1. Introduction

Animal husbandry is a major industry in China, and mutton is the second highest in terms of meat production in this country, but the development and utilisation of sheep by-products is insufficient. Lamb tripe is nutritionally rich, with a protein content of 12.2%, a fat content of 3.4%, and high levels of minerals and vitamins, hence it is an emerging ‘health food’ and is of great practical significance to developing mutton by-products [1].

Oxidation and microbial contamination are the main factors leading to spoilage of meat and meat products during storage [2]. Numerous studies on the inhibitory mechanism of lipid oxidation in meat products have been carried out [3]. However, few studies on the control of protein oxidation in meat products have been reported. Protein oxidation plays an essential role during the processing and storage of meat products and can have an unpredictable effect on meat safety and deterioration [4]. Therefore, it is necessary to find effective methods to control protein oxidation during the processing and storage of meat and meat products.

Grape seed extracts (GSEs) are widely used in the food industry, have generally recognised as safe (GRAS) status, and can inhibit free radical generation [5,6]. GSEs are mostly monomeric flavan-3-ols and oligomeric grape seed proanthocyanidins (GSPs) [6,7,8,9]. GSPs are effectively scavenge superoxide and hydroxyl radicals. The antioxidant activity of GSEs is affected by the degree of polymerisation, with greater polymerisation generally equating to higher antioxidant activity [10,11,12,13]. The antioxidant effects of these polyphenols are dependent on the biological activity, stability, and bioavailability of the active ingredients. However, GSEs are insoluble in water, readily decompose when exposed to light, and undergo moisture absorption and hydrolysis. These phenomena can be remedied by inclusion of β-cyclodextrins (β-CDs), cyclic oligosaccharides produced by enzymatic degradation of starch that received GRAS status designated by the U.S. Food and Drug Administration (FDA) in 1998 [14]. β-CDs have a truncated cone-shaped structure with a hydrophilic outer surface and a lipophilic inner cavity, and these physicochemical properties allow them to capture guest molecules in their cavities [15,16,17,18]. HP-β-CD is modified β-CDs containing seven D-glucopyranoside fragments of hydroxypropyl in the sixth position, resulting in better solubility in water [19,20]. Researchers have studied various applications of HP-β-CD, including encapsulation of polyphenols [1,21,22]. For example, polyphenol-cyclodextrin inclusion complexes not only prevent dityrosine cross-linkage, but also reduce protein oligomerisation [23], and they can reduce the partial degradation of essential amino acids, thereby decreasing adverse interactions between proteins and polyphenols [24]. Thus, the naturally unstable antioxidant improves antioxidant effects and bioavailability through inclusion. Thus, effective methods for linking HP-β-CD/polyphenol inclusion complexes and proteins needs are in demand. Molecular docking has emerged as a useful tool for bioinformatics and computational chemistry analyses, and it can provide information on ligand binding to receptors [25].

In this paper, hydroxypropyl cyclodextrin was considered as the host molecule and grape seed extract as the guest molecule. The inclusion complex was prepared by the method of co-precipitation in different host–guest ratios. The effect, structure, and properties of the inclusion complex prepared in different proportions were discussed and then the application of inclusion complex in controlling myofibrillar protein oxidation has also been studied.

## 2. Results and Discussion

### 2.1. Encapsulation Efficiency and Loading Capacity

Table 1 shows the encapsulation efficiency and loading capacity of HP-β-CD/GSE inclusion complexes with different molar ratios prepared by co-precipitation method. When the molar ratio of HP-β-CD/GSE inclusion complexes increased from 1:0.5 to 1:1, the encapsulation efficiency of the inclusion complexes decreased from 63.2 ± 3.31% to 26.5 ± 2.54%. The encapsulation efficiency and loading capacity of HP-β-CD/GSE inclusion complexes increased to 40.7 ± 1.33% and 4.60 ± 0.37% at the molar ratio of 1:2 respectively comparing to the molar ratio of 1:1. The results indicated that the HP-β-CD/GSE inclusion complex had the best effect at the molar ratio of 1:0.5. This probably occurs as the hydrophobic cavity of HP-β-CD molecule had enough space to encapsulate and interact with GSE through hydrogen bonding and Van der Waals forces [26]. The encapsulation efficiency of inclusion complexes was improved at the molar ratio of 1:2, which is maybe due to GSE’s ability to adhere to the surface of the inclusion complex.

### 2.2. FTIR Spectra

FT-IR spectroscopy is a useful technique to identify which vibrational modes of the molecules are being disrupted during the inclusion process, suggesting changes in the characteristic bonds of the guest molecule [27]. In this way, inclusion complexes can be demonstrated by study of the modification of the peak shape, position, and intensity [28]. GSE exhibited a shallow broad absorption peak at 3431 cm^−1^ in Figure 1, which may be due to the stretching vibration of O–H bond in the phenolic hydroxyl group, and the characteristic skeleton vibrations are mainly concentrated around 1000–1650 cm^−1^ and 700–850 cm^−1^ regions. Due to the existence of an O–H stretching vibration, the FTIR spectra of HP-β-CD include significant absorption bands at 3416.08 cm^−1^, C–H stretching vibrations at 2928.33 cm^−1^, and symmetrical and asymmetric C–O–C stretching vibrations at 1159 cm^−1^ and 1032 cm^−1^, respectively. It can be seen that there is a shallow absorption peak at 3431 cm^−1^ from f(GSE) in FTIR spectra. After formation of the HP-β-CD/GSE inclusion complex, the characteristic GSE peaks were reduced, shifted, or lost completely. The absorption peak of GSE corresponding to 3431 cm^−1^ shifted slightly to 3410 cm^−1^, and a new absorption peak appeared at 2988 cm^−1^. The C–O tensile vibration at 1000–1300 cm^−1^ was obscured by the characteristic HP-β-CD absorption peak and broad absorption peaks of HP-β-CD appeared in the FTIR spectra of inclusion complexes, which indicated that GSE may have been completely embedded in the cavity of HP-β-CD. These changes may be caused by the formation of HP-β-CD/GSE inclusion complex. The spectra of physical mixture are a simple superposition between single components of HP-β-CD and GSE. The results of FTIR confirmed the formation of the HP-β-CD/GSE inclusion complex, while other methods provided further verification. The results were consistent with those of previous studies [29,30] in which HP-β-CD was used to encapsulate tertiary butylhydroquinone (TBHQ).

### 2.3. XRD Analysis

XRD is one of the most accurate methods for characterising the formation of inclusion complexes between HP-β-CD and guest molecules. The crystalline properties of a guest molecule can change upon forming an inclusion complex with HP-β-CD [25,31]. Figure 2 shows that the X-ray diffraction pattern of HP-β-CD has a broad diffraction peak at 2θ of 9.6° and 24.7°, confirming its amorphous structure [27]. However, GSE showed characteristic diffraction peaks at 3.4°, 8.2°, and 9.4° at 2θ due to its crystalline property. The patterns of different molar ratios of HP-β-CD/GSE inclusion complexes were substantially different from those of pure GSE and physical mixture and displayed broad diffraction peaks with decreased intensities as well as the complete absence of primary diffraction peaks corresponding to GSE, which revealed the changes of GSE from crystal to amorphous state. GSE lost crystalline property during its preparation by HP-β-CD inclusion and may exist in an amorphous form within the cavity of HP-β-CD. Additionally, these results were consistent with those of FTIR spectroscopy.

### 2.4. DSC Analysis

Figure 3 shows differential scanning calorimetry (DSC) curves of HP-β-CD, GSE, their physical mixtures, and their inclusion complexes. GSE exhibited a sharp endothermic melting peak at 238.21 °C. Meanwhile, HP-β-CD displayed a broad endothermic peak at 82.56 °C, related to moisture loss from HP-β-CD. A second endothermic melting peak of HP-β-CD was observed at 349.38 °C, which may be due to thermal decomposition. The DSC curve of the physical mixture was a simple superposition of the curves of HP-β-CD and GSE. This result indicates that the two substances were only physically mixed but did not interact with each other.

In HP-β-CD/GSE inclusion complexes with different molar ratios, except for the disappearance of GSE melting peaks at 238 °C, only the characteristic absorption peak of HP-β-CD/GSE (1:0.5) inclusion complex is evident in the DSC curve at 88 °C. There are two endothermic melting peaks in the DSC curve of HP-β-CD/GSE (1:1) inclusion complex, at 74.99 °C and 341.70 °C. There are two endothermic melting peaks in the DSC curve of HP-β-CD/GSE (1:2) inclusion complex, at 73.65 °C and 323.35 °C, which may be due to the formation of new intermolecular or intramolecular bonds between HP-β-CD and GSE during the formation of the inclusion complex. The melting point of GSE disappeared, and the thermal properties of HP-β-CD were altered following formation of the inclusion complex between GSE and HP-β-CD, resulting in a shift in the melting peak. DSC analysis provides exhaustive data related to physical changes [32] and can be used to determine the formation of solid inclusion complexes, and the disappearance of thermal peaks containing the guest molecule further indicates the successful preparation of the inclusion complex [25,31,33].

### 2.5. SEM Analysis

SEM can be used to determine surface morphology following interactions between electrons and substances, providing an auxiliary method to monitor the formation of inclusion complexes [34]. As shown in Figure 4, GSE displayed an acicular rhombic crystal morphology, while HP-β-CD presented a spherical shape. Both forms were observed in the physical mixture of HP-β-CD and GSE, visible as an irregular block structure and a fragmented spherical structure in which the forms of the host (HP-β-CD) and guest (GSE) were unchanged. The shape of the inclusion complex was completely different from the shape of the physical mixture, which showed a multi-layer block structure with the surface of spherical cavity or cavity fragment structure. The surface morphology of HP-β-CD/GSE inclusion complexes with different molar ratios differed; inclusion complexes of HP-β-CD/GSE (1:0.5) presented a round spherical block structure, while those at 1:1 and 1:2 ratios presented an irregular layered structure. It is widely reported in literature that upon inclusion of guest molecules within cyclodextrins (CDs), the surface of the inclusion complex changes greatly due to a loss in the crystallinity of guest molecules [29,35,36]. In HP-β-CD/GSE inclusion complexes, the original morphology of the two components was lost completely, small aggregates of irregular size were stacked to form a multilayer block structure. Together, the FTIR, XRD, DSC, and SEM results indicate the successful formation of the HP-β-CD/GSE inclusion complex.

### 2.6. DPPH Free Radical Scavenging Activity

The compound 2,2-diphenyl-1-picrylhydrazyl (DPPH) is a stable system that is widely used for evaluation of antioxidant-mediated removal of free radicals. The antioxidant capacity of phenolic compounds depends mainly on the positions of hydroxyl groups and the degree of hydroxylation. Figure 5 shows the effects of GSE on DPPH radical scavenging activity, which were ordered HP-β-CD/GSE (1:2) > HP-β-CD/GSE (1:1) > HP-β-CD/GSE (1:0.5) > GSE. This could be because catechins of the inclusion polyphenol and the two hydroxyl groups at the rim of β-CD engage in hydrogen bonding interactions, and the OH groups in the β-CD wall may be shielded by catechins, which stabilises them and makes them more resistant to oxidation [23]. The result was consistent with that of a previous study [37] in which the antioxidant activity of the rosmarinic acid (RA):HP-β-CD freeze-dried complex was higher than that of RA.

### 2.7. Molecular Docking

Figure 6a shows the possible conformation of the inclusion complex determined by molecular docking. The low affinity of GSE in the HP-β-CD/GSE complex was estimated to be −7.0 kcal/mol, high affinity between host and guest means high absolute value of the binding energy and thus more stable inclusion complex. There are two possible ways for GSE to enter the HP-β-CD cavity, with equal affinity. The supramolecular structure was maintained by intermolecular hydrogen bonds, consistent with the results of thermal analysis and FTIR spectroscopy.

Molecular docking is a modern theoretical analysis method that can provide information on the interaction mechanisms between small molecules and proteins. Binding of the HP-β-CD/GSE inclusion complex to myofibrillar protein (MP) was dominated by hydrogen bonding and Van der Waals interactions. As shown in Figure 6b, the predicted binding residues included GLN-60, ASP-57, GLU-94, ARG-20, PRO-28, and VAL-3. The HP-β-CD/GSE inclusion complex interacts with its target MP via hydrogen bonds (involvement of the ASP-57 and VAL-3 residues), ionic interactions (ARG-29, GLU-94, and GLN-60) and Van der Waals (PRO-28 and LU-215), resulting in an optimal binding energy of −9.42 kcal mol^−1^. Docking modelling is a useful tool for probing protein-ligand interactions and supporting experimental results. Appropriate molecular modelling studies can provide guidance for ligand screening and assist optimisation at specific sites.

### 2.8. Effect of HP-β-CD/GSE Inclusion Complex on the Oxidation of Myofibrillar Protein

Carbonyl content is generally considered an indicator of the degree of oxidation of a protein. In general, the higher the carbonyl content, the higher the degree of protein oxidation. As shown in Table 2, the carbonyl content was 1.19 nmol/mg before oxidation and increased to 2.23 nmol/mg after 12 h of oxidation. After adding the HP-β-CD/GSE inclusion complex, the carbonyl content decreased significantly, indicating effective inhibition of oxidation. This may be attributed to the fact that plant polyphenols act as chelating agents for transition metal ions, and as free radical scavengers that inhibit oxidative modification of proteins. Previous research [38] showed that the iron chelating agent caffeic acid can act as an antioxidant by scavenging reactive oxygen species (ROS) and free radicals, thereby reducing the amount of •OH radicals produced by Fenton reactions, thus reducing the oxidation of meat proteins.

The carbonyl content of MP in the presence of the HP-β-CD/GSE inclusion complex at a high dose (155 μmol/g protein) was higher than that in the low dose groups. A previous report [39] showed that low doses of polyphenols improve the gelling and emulsifying ability of MP, but high doses can cause aggregation, reduce solubility, and increase carbonyl content, resulting in adverse changes to MP structure and properties.

Many proteins contain numerous –SH groups that are sensitive to hydroxyl radicals and can be readily converted into disulfide bonds (-S-S-) within and between molecules, resulting in protein surface aggregation. Thus, the sulfhydryl content can also be used as an important indicator of protein oxidative denaturation. As shown in Table 2, the total sulfhydryl content was 58.33 nmol/mg after oxidation for 12 h, while the total sulfhydryl content of the HP-β-CD/GSE inclusion complex was 61.37 nmol/mg after adding the HP-β-CD/GSE inclusion complex at a dose of 155 μmol/g. An appropriate concentration of HP-β-CD/GSE inclusion complex reduced the sulfhydryl content of MP caused by oxidation, which may be due to the formation of mercaptan–quinone compounds.

Bromophenol blue (BPB) molecules can bind to hydrophobic binding sites on the surface of protein molecules, hence the amount of BPB bound can be used as an indicator of the hydrophobicity of the protein surface [40]. In general, an increase in surface hydrophobicity indicates protein unfolding, which is usually accompanied by altered functional properties.

### 2.9. SDS-PAGE Analysis

The effects of oxidation and addition of the HP-β-CD/GSE inclusion complex (1:2) on protein crosslinking were analysed by SDS-PAGE. As shown in Figure 7, compared with oxidised MP under reducing conditions, addition of the HP-β-CD/GSE inclusion complex (1:2) restored the myosin heavy chain (MHC) band, which indicates that protein polymerisation caused by oxidation mainly resulted from the cross-linking of MHC molecules via disulfide bonds. When the HP-β-CD/GSE inclusion complex was added at a final concentration of 5, 55, or 105 μmol/g, there were no significant effects on protein crosslinking or aggregation. However, when the content was 155 μmol/g, the actin band colour became less intense, which indicates that protein crosslinking and aggregation were promoted in the presence of higher concentrations of polyphenols, and actin also participated in the formation of macromolecular substances [41]. This may be because Epigallocatechin gallate in the higher concentrations of HP-β-CD/GSE inclusion complex is oxidised to corresponding quinone derivatives that subsequently react with mercaptans to form thiol-quinone adducts. Quinone derivatives can be used as protein cross-linking agents to produce protein polymers [39].

## 3. Material and Methods

### 3.1. Materials

The fresh lamb tripe was randomly purchased from Xinjiang Western Animal Husbandry Co., Ltd. (Shihezi, China). Grape seed extract (OPC, 95%) was purchased from Aladdin (Shanghai Aladdin Biochemical Technology Co., Ltd, Shanghai, China). HP-β-Cyclodextrin (Mw 1541.54), ferric chloride, ascorbic acid, hydrogen peroxide (30%), ethylenediamine tetraacetic acid, 2,4-dinitrophenylhydrazine (DNPH), 5,5′-Dithiobis (2-nitrobenzoic acid), Sodium dodecyl sulfate and β-mercaptoethanol, bromophenol blue (BPB) were purchased from Sigma−Aldrich (Sigma−Aldrich, Co., St. Louis, MI, USA).

### 3.2. Methods

#### 3.2.1. Preparation of Inclusion Complexes (IC)

Inclusion complexes were prepared as described previously [42,43,44] with slight modifications. HP-β-CD:GSE complexes were prepared at molar ratios of 1:0.5, 1:1, and 1:2. After HP-β-CD was fully dissolved in 40 mL of 30% ethanol in a conical flask wrapped in an aluminium foil cone, polyphenols were added to achieve various final concentrations. Solutions were shaken at 40 °C for 72 h prior to rotary evaporation and placed in a refrigerator at 4 °C overnight. The sediments were obtained by vacuum filtration and washed with ethanol, then frozen at −80 °C for 24 h and freeze-dried in a freeze dryer for 12 h to obtain the resulting solid complexes.

Mixtures of HP-β-CD/GSE were prepared by mixing HP-β-CD with GSE to homogeneity at a molar ratio of 1:1 prior to fully grinding and mixing in a mortar.

#### 3.2.2. Determination of Encapsulation Efficiency and Loading Capacity

An accurate 10 mg of inclusion complex was dissolved in 10 mL ethanol. After ultrasonic treatment for 10 min, the solution was centrifuged at 2000 rpm for 5 min, then diluted according to a certain proportion. The absorbance was measured at 550 nm with a microplate reader. The concentration of GSE was calculated according to the standard curve, and the calculation formula of encapsulation efficiency was as follows:$$EE\%=\frac{A}{A_{0}}\times 100{\%}_{\;}LC\%=\frac{A}{A_{1}}\times 100{\%}_{\;}$$
where *A* is the content of GSE in the inclusion complex, *A*_0_ refers to the total amount of GSE, and *A*_1_ is the mass of inclusion complex.

#### 3.2.3. Fourier-Transform Infrared Spectroscopy (FTIR)

FTIR was performed as described previously [45]. FTIR absorption spectra of HP-β-CD/GSE and the physical mixture of polyphenol and HP-β-CD were separately analysed with potassium bromide using the press pellet technique. Samples were scanned over a range of 4000−400 cm^−1^ using an FTIR spectrometer (U.S. Seamus Nicolet IS10, Shihezi University, Xinjiang, China).

#### 3.2.4. X-Ray Diffraction (XRD)

An appropriate amount of HP-β-CD/GSE, and the physical mixture of polyphenol and HP-β-CD, were separately evenly placed on a Cu Kα1/graphite monochromator, and XRD data were collected at a 40 kV tube voltage and 40 mA tube current covering a 2θ angular range of 10–90° with a step size of 0.02°.

#### 3.2.5. Differential Scanning Calorimetry (DSC)

DSC was performed as described previously [41] with slight modifications. An appropriate amount of sample was measured using a DSC (model 200F3, Hangzhou, Zhejiang, China). Samples were scanned over a range of 30–400 °C with a heating rate of 10 °C/min and a dry nitrogen flow rate of 25 mL/min.

#### 3.2.6. Scanning Electron Microscopy (SEM)

SEM was performed as described previously [46]. Coated samples of HP-β-CD/GSE and the physical mixture were produced according to the operation requirements of the scanning electron microscopy (model JSM-6490LV,J, Jiangnan University, Wuxi, China). The surface morphology was analysed using SEM and images were captured at an excitation voltage of 10 kV.

#### 3.2.7. 2,2-diphenyl-1-picrylhydrazyl (DPPH) Free Radical Scavenging Activity

Samples were prepared at different concentrations (0–1.0 mg/mL) in aqueous solution, and 0.2 mmol/L DPPH solution was prepared with anhydrous ethanol and stored away from light. Sample solution (1 mL) and anhydrous ethanol solution (1 mL) were mixed and the absorbance at 517 nm was designated *A*_0_. Sample solution (1 mL) and DPPH solution (1 mL) were mixed and the absorbance at 517 nm was designated *A*_1_. Anhydrous ethanol (1 mL) and sample solution (1 mL) were mixed and the absorbance at 517 nm was designated *A*_2_. After incubating in the dark at room temperature for 30 min, 96-well plates were scanned at 517 nm using a microplate reader. Sample solution was replaced with deionised water as a blank control. The percentage antioxidant activity (*AA*%) was then calculated using the formula:$$AA\%=\left(1-\frac{A_{1}-A_{2}}{A_{0}}\right)\times 100$$

#### 3.2.8. Molecular Docking

In order to investigate the interactions between HP-β-CD and GSE, molecular docking was performed using Autodock Tools program (Version 1.5.6, Scripps Institute, La Jolla, CA, USA) [47]. The 3D structure of HP-β-CD was retrieved from the PubChem database (Pubchem CID: 14049689). The molecular structure of HP-β-CD was obtained by modifying specific hydroxyl groups, and the energy optimisation was performed. The 3D structure of GSE was obtained from the ChemSpider database (ChemSpider ID: 32079101), plotted with chemdraw, loaded into ChembioDraw Ultra (Version 1.3.0, Cambridgesoft corporation, Cambridge, MA, USA), and then structurally optimised with MM2 force field. In order to identify all binding sites in GSE, Autodock Tools were used to process the molecular structures of GSE and HP-β-CD, allocate atomic types and partial charges. HP-β-CD acted as a receptor, and GSE acted as a ligand. The grid spacing of the docking box was 1 Å, and the entire receptor was covered by setting X, Y, and Z centres to 0.126, 0.662, and 0.109, and x, y and z arg sizes to 16, 14, and 14 in the HP-β-CD box. Grid maps of various atoms of ligands and receptors were generated using AutoGrid (Version 5.9.1, AuiltoGrid systems, Inc., Redwood City, CA, USA), and 200 flexible ligand docking modellings were performed to search for possible docking points and corresponding energies after grid map generation [48]. During the docking process, the exhaustiveness parameter was set to 32, 9 conformations can be generated, and the other parameters were default. The optimal conformation was chosen as the final docking conformation for visual analysis by PyMol (Version1.7.0, Schrodinger LLC., New York, NY, USA).

#### 3.2.9. Extraction of Myofibrillar Protein (MP) and MALDI-TOF-MS Analysis

MP from lamb tripe was extracted as described previously [49] with minor modifications. A 5 g sample of treated lamb tripe was homogenised in phosphate buffer (0.1 M NaCl, 2 mM MgCl, 1 mM EGTA, pH 7.0), centrifuged, and the supernatant was discarded. The obtained precipitate was mixed with NaCl (0.1 M, pH 6.25) and filtered through four layers of gauze.

#### 3.2.10. Molecular Modelling Between Inclusion Complexes and MP

The docking between inclusion complex and MP was performed using the same software and procedure in 3.2.8. MP acted as a receptor, and the inclusion complex acted as a ligand. MP bands on sodium dodecyl sulphate-polyacrylamide gel electrophoresis (SDS-PAGE) gels (70 kDa) were excised and placed in Eppendorf tubes. The target sequence was 99% similar to the smooth muscle G actin DNase I complex (PDB ID: 3W3D) based on MALDI-TOF-MS analysis. Therefore, this crystal structure was used as a template for homology modelling to generate the 3D structure of the target sequence using Modeller (Version 9.18, Andrej sali lab., San Francisco, CA, USA). A total of 10 candidate models were constructed, and the structure with the best DOPE score was selected as the final model.

#### 3.2.11. Determination of Carbonyl Content

Different volumes of HP-β-CD/GSE inclusion complex at molar ratios of 1:2 (0, 5, 55, 105, and 155/g protein) dissolved in 0.15 M phosphate buffer were separately prepared, Fenton solution (0.1 mM FeCl_3_, 0.1 mM V_C_, 20 mM H_2_O_2_) was added after shaking vigorously, and solutions were oxidised at 4 °C for 12 h. Protein solution without the HP-β-CD/GSE inclusion complex served as a non-oxidised control. DNPH derivatisation was used to estimate the carbonyl content of MP as described previously [50].

#### 3.2.12. Determination of Total Sulfhydryl Content

Briefly 8 mL of TRIS-glycine was mixed with 2 mL of treated MP solution. After centrifugation, 0.5 mL of 10 mM Ellman’s reagent was added and the absorbance at 412 nm was measured after reacting for 30 min using a microplate reader. A control lacking MP solution was also included. A molar extinction coefficient of 13,600 M^−1^ cm^−1^ was used to calculate the total sulfhydryl content.

#### 3.2.13. Determination of Surface Hydrophobicity

A 200 μL of 1 mg/mL Bromophenol Blue (BPB) was added to 1 mL of each MP sample, mixed, and centrifuged at 5000 rpm for 15 min. The absorbance of the supernatant at 595 nm was measured and denoted as A. A blank sample BPB with phosphate buffer was denoted as A_0_. Surface hydrophobicity was then calculated using the following formula:$$BBPbound\left(\mu g\right)=\frac{200\mu g\times \left(A_{0}-A\right)}{A_{0}}$$

## 4. Conclusion

Three HP-β-CD/GSE inclusion complexes with host-guest ratios of 1:0.5, 1:1 and 1:2 were successfully prepared by co-precipitation method. The encapsulation efficiency and loading capacity were the most optimal when the molar ratio was 1:0.5. Fourier transform infrared spectroscopy (FTIR), X-ray diffraction (XRD), differential scanning calorimetry (DSC) and scanning electron microscopy (SEM) were used to characterise the three inclusions. In the meantime, the three-dimensional supramolecular structure of HP-β-CD/GSE inclusion complex was successfully simulated by molecular docking. Then the MP bands were analysed by MALDI-TOF-MS and compared in PDB protein structure database. The consistency between the MP bands and the target sequence of smooth muscle G actin DNase I complex (PDB ID:3W3D) was 99%. The structure was used as a template and homology modelling was carried out to construct the three-dimensional structure of target sequence. Moreover, the antioxidant activity of the inclusion complex was discussed. The results showed that the DPPH radical scavenging effect was the most optimal when the molar ratio was 1:2, which was applied in MP oxidation system with 20 mmol/L of H_2_O_2_. The antioxidant activity of the HP-β-CD/GSE inclusion complex at three different molar ratios was increased to different degrees; the HP-β-CD/GSE (1:2) inclusion complex displayed the best scavenging activity of DPPH-derived free radicals, and inhibited MP oxidation at low (5 μmol/g) to medium (105 μmol/g) concentrations, but significantly promoted MP oxidation at 155 μmol/g.

## Figures and Tables

**Figure 1 molecules-24-04487-f001:**
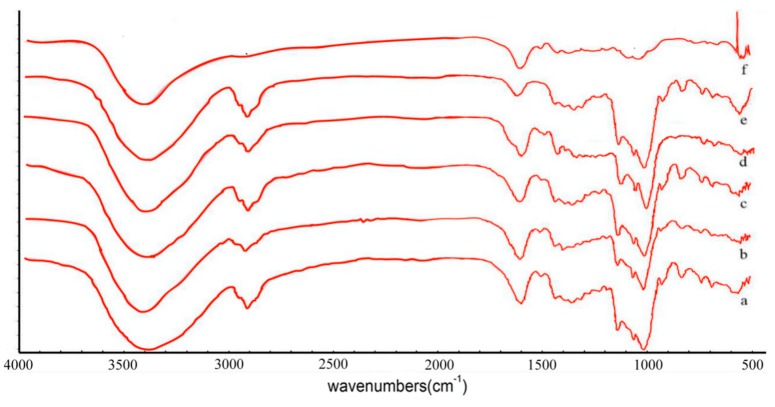
FTIR spectra of (**a**) HP-β-CD/GSE (1:0.5), (**b**) HP-β-CD/GSE (1:1), (**c**) HP-β-CD/GSE (1:2), (**d**) the physical mixture of HP-β-CD and GSE, (**e**) HP-β-CD, and (**f**) grape seed extract (GSE).

**Figure 2 molecules-24-04487-f002:**
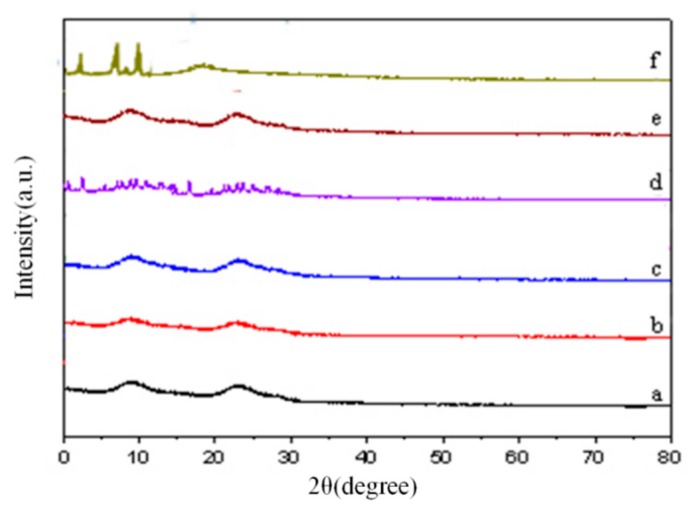
XRD of (**a**) HP-β-CD/GSE (1:0.5), (**b**) HP-β-CD/GSE (1:1), (**c**) HP-β-CD/GSE (1:2), (**d**) the physical mixture of HP-β-CD and GSE, (**e**) HP-β-CD, and (**f**) GSE.

**Figure 3 molecules-24-04487-f003:**
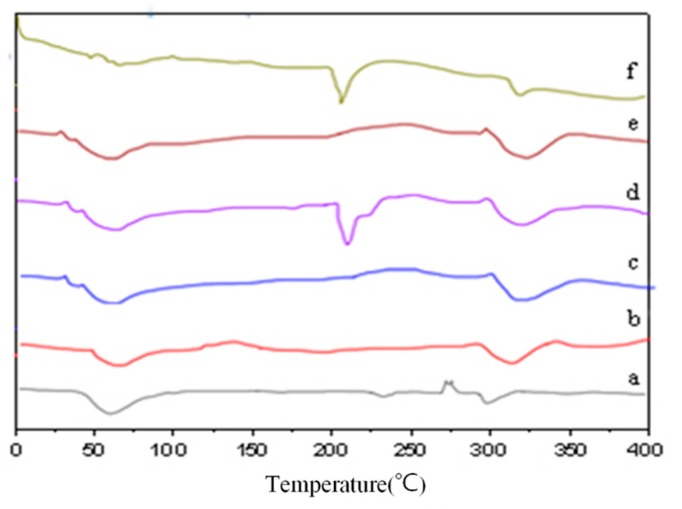
DSC of (**a**) HP-β-CD/GSE (1:0.5), (**b**) HP-β-CD/GSE (1:1), (**c**) HP-β-CD/GSE (1:2), (**d**) the physical mixture of HP-β-CD and GSE, (**e**) HP-β-CD, and (**f**) GSE.

**Figure 4 molecules-24-04487-f004:**
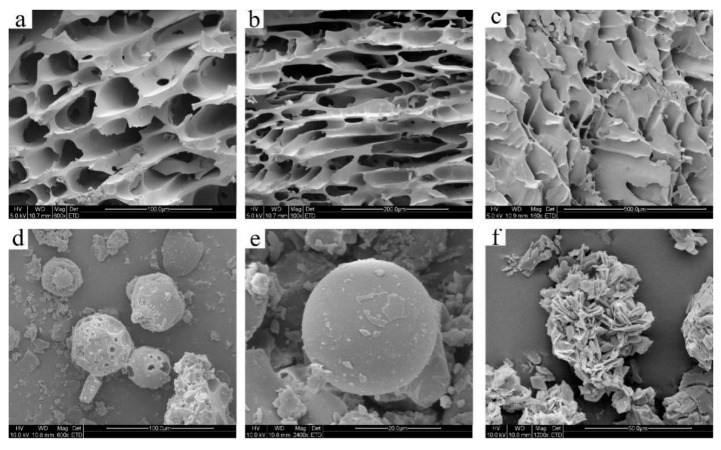
SEM of (**a**) HP-β-CD/GSE (1:0.5), (**b**) HP-β-CD/GSE (1:1), (**c**) HP-β-CD/GSE (1:2), (**d**) the physical mixture of HP-β-CD and GSE, (**e**) HP-β-CD, and (**f**) GSE.

**Figure 5 molecules-24-04487-f005:**
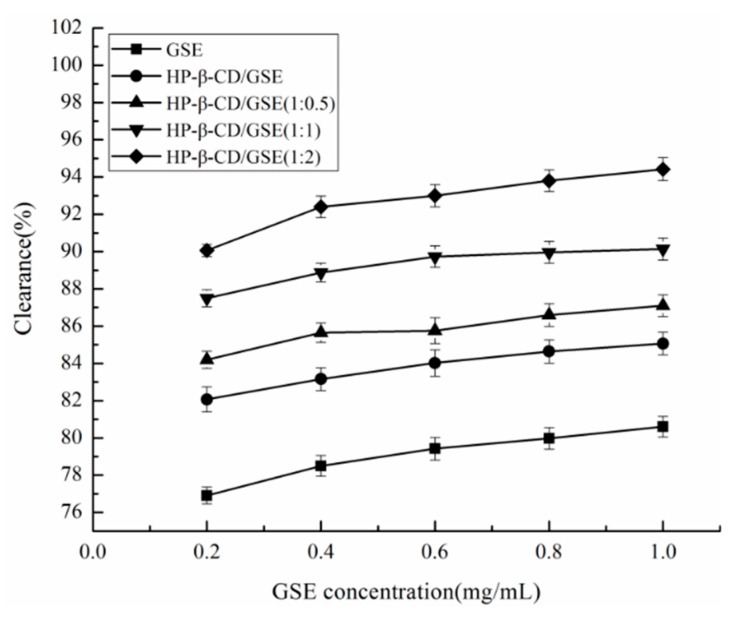
The DPPH radical scavenging ability of GSE and the HP-β-CD/GSE inclusion complex at different molar ratios.

**Figure 6 molecules-24-04487-f006:**
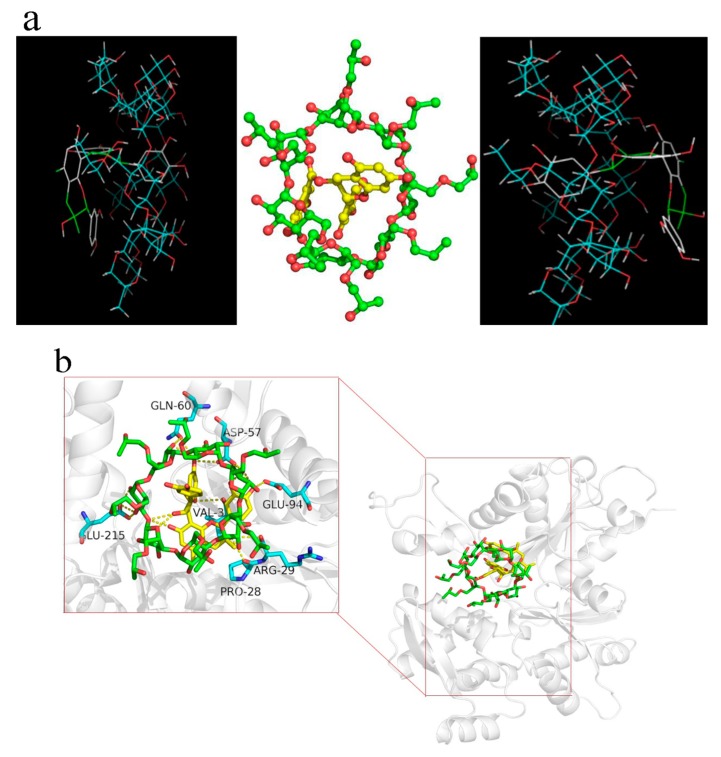
(**a**) Molecular docking of the HP-β-CD/GSE inclusion complex (1:2). (**b**) Predicted interactions between the HP-β-CD/GSE inclusion complex (1:2) and myofibrillar protein (MP).

**Figure 7 molecules-24-04487-f007:**
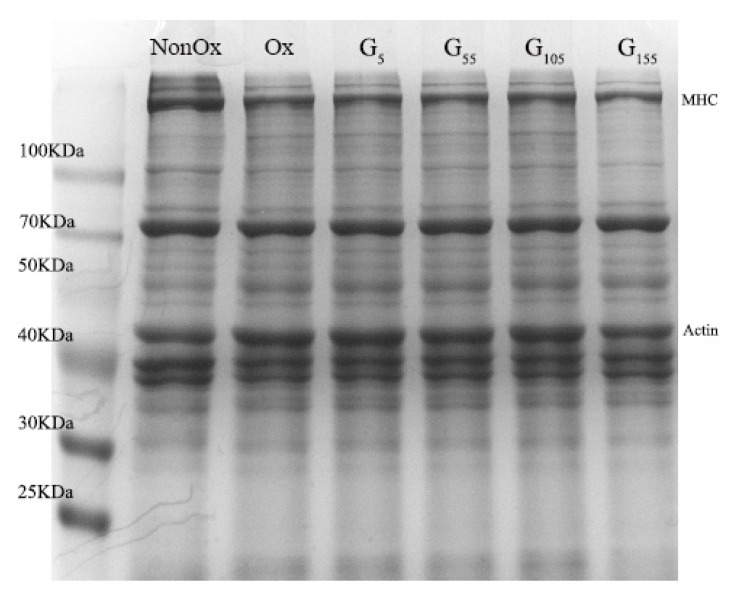
SDS-PAGE patterns of myofibrillar protein samples. NonOx: Unoxidised MP Sample; Ox: Oxidised MP Samples; G: HP-β-CD/GSE inclusion complex (5, 55, 105, 155 μmol/g) were added at different concentrations under oxidation conditions.

**Table 1 molecules-24-04487-t001:** Encapsulation efficiency and loading capacity of HP-β-CD/GSE inclusion complex.

HP-β-CD/GSE Ratio	Encapsulation Efficiency	Loading Capacity
1:0.5	63.2 ± 3.31%	5.41 ± 0.24%
1:1	26.5 ± 2.54%	2.59 ± 0.13%
1:2	40.7 ± 1.33%	4.60 ± 0.37%

**Table 2 molecules-24-04487-t002:** Changes of physicochemical indexes of myofibrillar proteins in different concentrations of HP-β-CD/GSE (1:2) treatment group.

Sample	Carbonyl Content (nmol/mg Protein)	Sulphydryl Content (nmol/mg Protein)	BPB Binding (μg)
NonOx	1.19 ± 0.12 ^a^	67.54 ± 0.78 ^a^	2.93 ± 1.15 ^e^
Ox	2.23 ± 0.22 ^c^	58.33 ± 1.16 ^e^	7.16 ± 1.94 ^a^
Ox + HP-β-CD/GSE (5 μmol/g)	1.86 ± 0.56 ^bc^	60.02 ± 1.97 ^d^	6.04 ± 0.37 ^b^
Ox + HP-β-CD/GSE (55 μmol/g)	1.31 ± 0.17 ^ab^	61.43 ± 2.06 ^c^	5.89 ± 2.34 ^c^
Ox + HP-β-CD/GSE (105 μmol/g)	1.26 ± 0.09 ^a^	62.21 ± 1.93 ^b^	4.96 ± 1.71 ^d^
Ox + HP-β-CD/GSE (155 μmol/g)	1.43 ± 0.38 ^ab^	61.37 ± 1.85 ^c^	6.07 ± 0.83 ^b^

Notes: Values are means ± standard deviation (SD). Different lowercase letters indicate significant differences (*p* < 0.05). NonOx (unoxidised MP), Ox (oxidised MP), Ox + HP-β-CD/GSE, and HP-β-CD/GSE inclusion complex (5, 55, 105, 155 μmol/g) samples were separately added at different concentrations under oxidative conditions.

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
