# Peer review of "Preparation and Characterisation of Polyphenol-HP-β-Cyclodextrin Inclusion Complex that Protects Lamb Tripe Protein against Oxidation"

_molecules, 2019, doi:10.3390/molecules24244487_

Round 1
Reviewer 1 Report
The revised manuscript by Li et. al. describes the characterization of the inclusion compound of GSE/HP-b-CD by various methods and its impact on the oxidation of the myofibrillar protein from lamb tripe.
Only some minor issues need reconsideration.
line 18: “…three-dimensional supramolecular structure of inclusion complex of…” to “…three-dimensional supramolecular structure of the inclusion complex of…”
line 20: “…PubChem protein structure database...” to “ …chemical database PubChem and structural database PDB…”
line 81: “Table 1 showed…” to “Table 1 shows…”
line 195: Figure 6. Use “a)” and “b)” (lowercase) instead of “A)” an “B)”. Legend should be more descriptive. A verbose description of the depicted molecules should be given.
line 207: GLV-94 to GLU-94
lines 208-211: “HP-β-CD/GSE inclusion complexes interacted with ASP-57 and VAL-3 of MP through hydrogen bonds, and were stabilized by ARG-29, GLU-94 and GLN-60 through ionic interactions PRO-28 and LU-215 interacted with CS through Van der Waals forces, and the optimal binding energy was -9.42 kcal mol-1” to “The HP-β-CD/GSE inclusion complex interacts with its target MP via hydrogen bonds (involvement of the ASP-57 and VAL-3 residues), ionic interactions (ARG-29, GLU-94 and GLN-60) and Van der Waals forces (PRO-28 and GLU-215) resulting in an optimal binding energy of -9.42 kcal mol-1”
line 207: “OPC” to “GSE”
line 355: “Kda” to “kDa”
line 359: “…modeler 9.18 software” to “…Modeller v9.18 software”
Authors must cite the programs that the used in their research (Pymol, Modeller, Autodock Vina, etc.)
The abbreviation for b-CD can be omitted from the abbreviation table.
Reviewer 2 Report
The manuscript should be improved in order to fit the quality standards of the journal.
There are some observations to the paper.
The state of the art must be improved, there is not enough references
A better discussion should provide about the results of Encapsulation Efficient
In FTIR there are new signals or lost signals where a deep explanation should be given, in line 98 it is mention that there is a new signal at 2988 cm-1, and claim completely embedded GSE into the HPCD cavity but the loss of signals not necessary means is inside the cavity. Please explain to what this new signal s are attributed.
XRD peaks are not indexed, the authors attributed that GSE peaks disappeared because the GSE changes from crystal to amorphous structure. Could you explain how this happen.
In DSC there are two peaks between 270-280 C for the 1:0.5 ratio, to what is attributed? And why only appears in the 1:0.5 ratio.
In figure 5 the DPPH free radical scavenging activity is shown but according to table 1, the highest loading capacity is for the 1:05 ratio, followed by 1:2 and 1:1, and the DPPH free radical scavenging activity is for 1:2, followed 1:1 and 1:05. These results can be contradictory, or the amount of GSE has not influence? Or what is the mechanism that make ration with less “loading capacity” to have the best results in DPPH free radical scavenging activity.
Round 2
Reviewer 2 Report
The authors have improved the manuscript and attend the suggestions, Now the article can be published in present form.
This manuscript is a resubmission of an earlier submission. The following is a list of the peer review reports and author responses from that submission.
Round 1
Reviewer 1 Report
The manuscript by Wenhui Li et al. regards a deep characterization of HP-β-CD/GSE inclusion complexes, prepared by co-precipitation method in different host-guest ratios, together with an interesting study on the application of this inclusion complex in controlling myofibrillar protein oxidation.
It is a well performed study, the results are innovative and the choice of the journal is appropriate. The paper is, from a general point of view well presented. Nevertheless, the following points need to be clarified before the paper can be accepted.
1) Page 2 line 60. Due to the relevance of the treated topic only two references (Ref. [14, 15]) are too few. More reference are suggested, such as:
- Ferrati, S.; Nicolov, E.; Bansal, S.; Hosali, S.; Landis, M.; Grattoni, A. Docetaxel/2-hydroxypropyl β-cyclodextrin inclusion complex increases docetaxel solubility and release from a nanochannel drug delivery system. Curr. Drug Targets 2015, 16, 1645-1649.
- Stancanelli, R.; Ficarra, R.; Cannavà, C.; Guardo, M.; Calabrò, M.L.;Ficarra, P.; Ottanà, R.; Maccari, R.; Crupi, V.; Majolino, D.; Venuti, V. UV–vis and FTIR-ATR characterization of 9-fluorenon-2-carboxyester/(2-hydroxypropyl)-β-cyclodextrin inclusion complex. J. Pharm. Biomed. Anal. 2008, 47, 704-709.
2) Please, for a better understanding of the text, put the "Materials and Methods" section before the "Results and Discussion" section.
3) Page 2 lines 80-84. These results are already reported in Table 1. They don't need to be reported also in the text. Please remove them from the text.
4) Page 3 lines 94-107. Please include in the text a reference to Fig. 1.
5) Page 3 lines 96-99. Actually, the absorption bands in the FTIR spectra at 2928.33 cm-1, 1159 cm-1 and 1032 cm-1 are not "due to the existence of an O-H stretching vibration". Please clarify this point.
6) Page 3 lines 100-101. This shift is not evident from the figure. Please put into evidence, in the figure, the absorption peak at 3431 cm-1 in the GSE spectrum, and the corresponding shift passing from physical mixture to inclusion complex.
7) Page 3 lines 103-104. This sentence is not clear. Please clarify it.
8) Page 4 lines 112-122. Please include in the text a reference to Fig. 2.
9) Page 4 lines 114-115 and lines 116-117. These two sentences seem to be contradictory. It would be better writing something like "due to the amorphous structure of HP-β-CD, its X-ray diffraction pattern has only broad diffraction peaks at..."
10) Page 4 lines 117-119. Please rewrite this sentence in order to clarify it.
11) Page 4 line 120. Please change in "...from crystal to amorphous state"
12) Page 4 line 135. -238°C?? Please correct.
13) Page 4 lines 134-140. These sentences must be rewritten. Their meaning is, in this way, misleading. In particular, please use "complex" and not "complexes" when you discuss each single case.
14) Page 5 lines 160-162. The observation, by SEM, of the morphological changes upon complexation, is a well known and widely reported method to verify complexation, as the authors wrote at the beginning of the paragraph. Hence, instead of "A previous study showed that" use "It is widely reported in literature that" and add some references, such as:
- Rajamohan, R.; Kothainayaki, S.; Swaminatan, M. Spectrofluorimetricstudy on inclusion complexation of 2-amino-6-fluorobenzothiazole with β-cyclodextrin. Collect. Czechoslov. Chem. Commun. 2008, 73, 147–160
- Venuti, V.; Crupi, V.; Fazio, B.; Majolino, D.; Acri, G.; Testagrossa, B.; Stancanelli, R.; De Gaetano, F.; Gagliardi, A.; Paolino, D.; Floresta, G.; Pistarà, V.; Rescifina, A.; Ventura, C.A. Physicochemical Characterization and Antioxidant Activity Evaluation of Idebenone/Hydroxypropyl-β-Cyclodextrin
Inclusion Complex†. Biomolecules 2019, 9, 531.
15) Page 5 line 164. "...strong evidence of a crystalline form". Please clarify this sentence, because HP-β-CD/GSE inclusion complex is in amorphous state.
16) Page 6 line 179. The title of the paragraph should be shortened.
17) Page 7 line 184-199. Please include in the text a reference to Fig. 6.
18) Page 8 line 225. Should it be Table 2?
Reviewer 2 Report
Dear Editor and Authors, No standardization of grape seed extract for the content of active compounds was carried out in the conducted tests. In view of the above, the authors did not carry out in sufficient detail:
- identity testing of the cyclodextrin complex,
- assessment of changes in the content of individual active compounds in the extract under oxidative conditions,
- docking inside cyclodestrin.
I believe that the authors should repeat the study, starting with the standardization of the extract and then repeating the next stages of the study.
Reviewer 3 Report
I reviewed the manuscript by Li et al., in which authors describe:
the characterization of the inclusion compound of GSE/HP-b-CD by various methods (FT-IR, XRD, DSC, etc.) the impact of the given complex on the oxidation of the myofibrillar protein from lamb tripe
Firstly, all the experiments were carefully designed, conducted and presented throughout the manuscript. However, it’s not clear throughout the whole manuscript for the reader whenever authors are referring to GSE as a crude oil mixture or as a single substance. Moreover, in paragraph 3.2.10 authors dedicate too much space and detail in the experimental part rather than in the Molecular Dynamics (MD) simulations (system preparation, protocol for minimization and MD runs). Actually, it is not clear whether authors conducted MD simulations or not, as they provide no results from such studies.
In general, I would like to recommend the publication of this manuscript in Molecules only after the following major issues are addressed:
In the Abstract section:
lines 12-13: “Grape seed extract (GSE) displays strong antioxidant activity, but instability a barrier to applications” should change to
“Grape seed extract (GSE) displays strong antioxidant activity, but its instability creates barriers to its applications”
line 17: “The Autodock Tools 1.5.6 was performed to simulate ...” should change to
“The Autodock Tools 1.5.6 was used to simulate …”
lines 21-24: “The binding mode of HP-β-CD/GSE inclusion complex and target protein was revealed from the molecular level, and the antioxidant ability of the resulting HP-β-CD/GSE inclusion complexes was investigated by measuring 2,2-diphenyl-1-picrylhydrazyl (DPPH) free radical scavenging” Authors should rephrase for clarity, i.e.
“The binding mode of the HP-β-CD/GSE inclusion complex to target protein was studied at the molecular level, and the antioxidant ability of the resulting HP-β-CD/GSE inclusion complexes was investigated by measuring 2,2-diphenyl-1-picrylhydrazyl (DPPH) free radical scavenging.
In the Introduction section:
lines 42-43: “Numerous studies on the inhibitory mechanism of lipid oxidation...” Authors should cite one or two relevant papers here.
line 50: “[6, 7, 5, 8]” Change the order to [5, 6, 7, 8]
lines 68-70: “Recently, ….” The word “Recently” should be removed as Molecular docking studies are not very new in the field of Biocomputing. One or two citations here can prove that.
line 71: “…hydroxypropyl cyclodextrin was used as the host molecule…” Change to “…hydroxypropyl cyclodextrin was considered as the host molecule …”
In the Results and discussion section:
lines 85-89: “This probably because the hydrophobic cavity of HP-β-CD had enough space to encapsulate and combine GSE through hydrogen bonding and Van der Waals force. The encapsulation efficiency of inclusion complexes was improved at the molar ratio of 1: 2, which maybe due to GSE adhering to the surface of the inclusion complex” should change to
“This probably occurs as the hydrophobic cavity of the HP-β-CD molecule had enough space to encapsulate and interact with GSE through hydrogen bonding and Van der Waals forces. The encapsulation efficiency of inclusion complexes was improved at the molar ratio of 1: 2, which is may be attributed to GSE’s ability to adhere to the surface of the inclusion complex”
A few notes here: Again, it’s not clear whether authors refer to GSE as a single substance or as the oil mixture. The predominant interactions in CD inclusion complexes are mainly the Van der Waals forces despite host-guest ratio. The interpretation of the produced results provided by authors here is not sufficient. Moreover, authors imply that there is an aggregation instead of inclusion complexing at the molar ratio of 1: 2. Are there any Phase solubility studies in the literature for that?
lines 175-178: “This could be because catechins of the inclusion polyphenol and the two hydroxyl groups at the rim of β-CD engage in hydrogen bonding interactions, and the OH groups in the β-CD wall may be shielded by catechins, which stabilises them and makes them more resistant to oxidation [20].”
Authors should rephrase for clarity, by using HP-β-CD instead of β-CD, specifying which CD rim (primary or secondary) are they referring to and providing more information about the degree of substitution of the HP-β-CD.
lines 184-186: “The lowest affinity of GSE in the HP-β-CD/GSE complex was estimated to be -7.0 kcal/mol, which was the most stable value among all inclusion complexes.”
Do author mean “The highest affinity of GSE in the HP-β-CD/GSE complex was estimated to be -7.0 kcal/mol, which was the lowest absolute value and thus the most stable case among all inclusion complexes”? A relative figure or a table with the rest values from their calculations may be needed here.
In the Material and methods section:
line 256: Authors should provide more information about the degree of substitution of the HP-β-CD. Did authors used any native β-CD in their experiments?
lines 326-332: Clearly, there is no molecular dynamics simulation here. Authors have performed some molecular modelling.
Reviewer 4 Report
It is interesting to use grape seed extract(GSE) for the industrial use. However, it is also quite important to keep the quality of GSE. The authors only describe GSE, the reviewer ask authors how many compounds are included in GSE ? The inclusion complex and physical mixture is completely different. The inclusion complex is heavily dependent on the molecular size of the guest compound. The molecular size of flavan flavonoid and also polyphenols is different, The authors should take much attention for these points. The experimental drawback is also found for the usage of KBr in FT-IR studies to clarify the inclusion complex. The mixing of the inclusion complex with KBr induces the destruction of the inclusion complex in most cases.
Based on these facts, the reviewer does not recommend this manuscript acceptable for publication in Molecules.
Round 2
Reviewer 2 Report
Dear Editor,
The authors did not carry out standardization of the raw material, therefore all of the tests performed were conducted as an approximation.
The results are not reliable from the point of view of the analytical methods used and the observed biological effects.
Reviewer 3 Report
I reviewed for second time the manuscript by Li et. al., with title “Preparation and Characterisation of Polyphenol-HP-b-Cyclodextrin Inclusion Complex that Protects Lamb Tripe Protein against Oxidation”.
Authors made a lot of effort to improve the presentation of their work and to enlighten point by point all the issues that I had previously raised.
However, authors still do not separate in the text the GSE crude oil mixture from the single substance GSE (may the use of different abbreviation could clarify things). Thus:
line 87-91: “This probably occurs as the hydrophobic cavity of HP-b-CD molecule had enough space to encapsulate and interact with GSE through hydrogen bonding and Van der Waals forces [26]. The encapsulation efficiency of inclusion complexes was improved at the molar ratio of 1: 2, which is maybe due to GSE’s ability to adhere to the surface of the inclusion complex.”
Authors should clarify here that they talk about a GSE mixture, 95% of which is procyanidins, based on the structure of GSE’s polyphenols, etc. Again, only assumptions can be made for host-guest interactions as authors are referring to the crude mixture.
lines 186-189: “This could be because catechins of the inclusion polyphenol and the two hydroxyl groups at the rim of b-CD engage in hydrogen bonding interactions, and the OH groups in the β-CD wall may be shielded by catechins, which stabilizes them and makes them more resistant to oxidation [23].”
Authors make also assumptions here.
line 271: As concern the Degree of Substitution (DS) of HP-b-CD
Indeed primary hydroxyl groups are situated on the narrow rim and secondary hydroxyl groups are situated on the wider rim of the cavities in CDs and HP-β-CD is a derivative of native b-CD, with substitutions of the -OH groups by hydroxypropyl groups, resulting in a dramatic improvement of its solubility and complexation ability than its parent b-CD. The query here is how many substitutions of the -OH groups by hydroxypropyl groups are present in the used chemical. The DS or approximate ~Mw of HP-b-CD should be given here.
As concern the computational studies:
lines 198-202: “The low affinity of GSE in the HP-β-CD/GSE complex was estimated to be -7.0 kcal/mol, which was the most stable value among all inclusion complexes. There are two possible ways for GSE to enter the HP-β-CD cavity, with equal affinity. The supramolecular structure was maintained by intermolecular hydrogen bonds, consistent with the results of thermal analysis and FTIR spectroscopy.”
Once again, high affinity between host and guest means high absolute value of the binding energy (more negative value) and thus more stable inclusion complex (not stable value). Modify accordingly.
Moreover, the title of paragraph 3.2.9 should change to “Extraction of myofibrillar protein (MP) and MS analysis” and include lines 342-344 in it. Title of paragraph 3.2.10 should change to “Molecular modelling between inclusion complexes and MP”
In general, I would suggest careful re-evaluation/removal of the computational section of the present paper. As the experimental work is novel and well conducted, a digested form of the paper may be more appropriate for the journal. I therefore believe that the paper as it stands is of insufficient impact for inclusion in Molecules, unless the major issues suggested above are addressed.
Reviewer 4 Report
It is quite difficult to understand the meaning of mixing CD with GSE. GSE is the mixture of many kinds of compounds. Some compounds might interact with CD, some do not. The reviewer can not understand the scientific meaning of this manuscript. There are huge studies of host-guest chemistry. In most cases, the experimental design is organized clearly using many techniques. Based on this line, the complex formation is determined as 1;1, 1:2, 2:2 2:3 and etc.The authors did not answer the question raised by the reviewer.